# Recovery Strategies in Endurance Athletes

**DOI:** 10.3390/jfmk7010022

**Published:** 2022-02-13

**Authors:** Robyn Braun-Trocchio, Austin J. Graybeal, Andreas Kreutzer, Elizabeth Warfield, Jessica Renteria, Kaitlyn Harrison, Ashlynn Williams, Kamiah Moss, Meena Shah

**Affiliations:** 1Department of Kinesiology, Texas Christian University, Fort Worth, TX 76129, USA; a.kreutzer@tcu.edu (A.K.); e.warfield@tcu.edu (E.W.); j.d.renteria@tcu.edu (J.R.); k.p.harrison@tcu.edu (K.H.); a.williams5@tcu.edu (A.W.); k.moss@tcu.edu (K.M.); m.shah@tcu.edu (M.S.); 2School of Kinesiology & Nutrition, University of Southern Mississippi, Hattiesburg, MS 39406, USA; austin.graybeal@usm.edu

**Keywords:** running, cycling, triathlons, recovery modalities, hydration, nutrition, sleep

## Abstract

In order to achieve optimal performance, endurance athletes need to implement a variety of recovery strategies that are specific to their training and competition. Recovery is a multidimensional process involving physiological, psychological, emotional, social, and behavioral aspects. The purpose of the study was to examine current implementation, beliefs, and sources of information associated with recovery strategies in endurance athletes. Participants included 264 self-identified endurance athletes (male = 122, female = 139) across 11 different sports including placing top three overall in competition (*n* = 55) and placing in the top three in their age group or division (*n* = 113) during the past year. Endurance athletes in the current study preferred hydration, nutrition, sleep, and rest in terms of use, belief, and effectiveness of the recovery strategy. Female endurance athletes use more recovery strategies for training than males (*p* = 0.043, *d* = 0.25), but not in competition (*p* = 0.137, *d* = 0.19). For training, top three finishers overall (*p* < 0.001, *d* = 0.61) and by division (*p* < 0.001, *d* = 0.57), used more recovery strategies than those placing outside the top three. Similar findings were reported for competition in top three finishers overall (*p* = 0.008, *d* = 0.41) and by division (*p* < 0.001, *d* = 0.45). These athletes are relying on the people around them such as coaches (48.3%) and fellow athletes (47.5%) along with websites (32.7%) for information and recommendations. Endurance athletes should be educated on other strategies to address the multidimensionality of recovery. These findings will be useful for healthcare professionals, practitioners, and coaches in understanding recovery strategies with endurance athletes.

## 1. Introduction

Participating in sport, whether at the professional or recreational level, is often a high demanding endeavor [1]. Endurance athletes traditionally undertake large training volumes to enhance adaptations and subsequently improve performance [2]. Endurance athletes place a great emphasis on their performance and spend a significant amount of time and money ensuring that they are in the best position to succeed in their respective sport, regardless of competition level.

Prioritizing high-level performance entails high training loads, often leading to an imbalance between stress and recovery [3]. Stress is defined as the destabilization or deviation from the norm in a biological and/or psychological system [4]. Physical stress refers to the physical fatigue induced during training and/or competition [5]. Physical stress with endurance athletes can be quantified in terms of training loads including external and internal training loads [6]. External training loads are objective measures of the work performed during training or competition (e.g., distance completed). Internal training loads assess the biological stress imposed on the athlete by the training or competition. Besides physical stress, other factors including psychological stress and mental fatigue influence an athlete’s ability to recover and readiness to train [7] Psychological stress occurs when people perceive that the demands from external situations were beyond their coping capacity [8]. An athlete with high emotional stress experiences irritation, aggression, anxiety, and inhibition [4], which can have a strong effect on the total stress and may negatively impact performance [9]. Therefore, endurance athletes need to monitor both physical and psychological stress.

Recovery is a multidimensional process involving a number of systems [9]. More specifically, recovery is defined as an inter- and intra-individual multi-level process for the re-establishment of performance abilities that can be systematically used to optimize situational conditions, to build up and refill personal resources, and buffers [4]. This definition demonstrates the complexity and multidisciplinary nature of the recovery process. Therefore, recovery programs should incorporate a variety of personalized and stressor-specific strategies focused on achieving a balance in psychosociophysiological components such as regaining physiological, psychological, emotional, social, and behavioral aspects of intensive training [4]. Recovery strategies can be divided into active and passive methods [3]; each strategy addresses either regenerative (the physiological aspect of recovery) or psychological components of recovery (mental and emotional stress) [5]. Active recovery strategies can consist of moderate exercise during the recovery process like walking, assisted squatting, sled dragging and pushing, mobility training, or de-loaded resistive training sessions. Passive recovery strategies include treatments such as massages, hot and cold baths, or just sitting or lying quietly.

Despite the importance and use of recovery strategies within athletics, specifically endurance athletes, recovery remains an under-researched area compared to training principles and competition strategies [10]. More specifically, the effects of gender on recovery have only recently gained research attention [11]. Due to sex differences in the physiological effects of exercise, it is important to customize recovery strategies accordingly. Female athletes benefit from using cooling recovery methods, such as cold-water immersion (CWI) due to their lower thermolytic capacities than males. Additionally, active recovery would be beneficial for female athletes since they have a greater decrease in arterial blood pressure after exercise compared to males. Furthermore, additional research is needed to examine differences in recovery strategies in athletes at varying skill levels to reduce the potential effects of overtraining syndrome [12], injury [1,13], mental fatigue [14], and impaired sleep [15]. Bezuglov and colleagues [16] reported that higher-tier endurance athletes utilize more recovery strategies, specifically massage, daytime nap, and sleep.

Currently, there is no clear protocol in terms of recovery strategies [5] and many athletes implement strategies based on personal experience rather than evidenced-based research [17]. Additionally, athletes may not be aware of the intended physical effects of a specific recovery strategy [18]. There is a need to understand the strategies that endurance athletes are utilizing and where they are receiving the information to enhance recovery from training and competition. Furthermore, few studies have examined the attitudes and beliefs of these recovery practices. For example, Murray and colleagues [10] showed that only 24% of the athletes utilized sleep as a recovery strategy despite it being rated and perceived as the most effective in Division I athletes suggesting a potential disconnect between athletes’ belief in the recovery strategy and their implementation. However, there are limited studies that have investigated this discord, particularly in endurance athletes. In order to extend and develop upon the existing literature, the purpose of the study was to examine current implementation, attitudes, beliefs, and sources of information associated with recovery strategies in endurance athletes.

## 2. Materials and Methods

### 2.1. Participants

Participants included 264 self-identified endurance athletes (male = 122, female = 139, non-binary = 1, preferred not to answer = 1, and did not answer = 1) across 11 different sports (cycling = 55, paracycling = 1, Nordic skiing = 1, running = 99, race walking = 1, rowing = 8, swimming = 7, triathlon = 89, wheelchair racing = 1, snowshoeing = 1, and aquabiking = 1) including professional athletes (*n* = 14), current collegiate athletes (*n* = 41), and former collegiate athletes (*n* = 42). The participants ranged in age from 18 to 79 years old (*M* = 39.82, *SD* = 13.84) and were primarily white (*n* = 231). Additionally, the majority of participants currently reside in the United States (*n* = 219). Placement in races during the past year included placing top three overall in competition (*n* = 55) and placing in the top three in their age group or division (*n* = 113).

### 2.2. Instruments

Participants completed an online questionnaire through Qualtrics that was based on the previous work of Murray et al. [19] and Murray et al. [10] to establish current practices, attitudes/beliefs towards recovery, and sources of information in endurance athletes. Modifications were made to the questionnaire to include additional recovery strategies along with additional questions regarding where the athletes receive their information regarding recovery strategies, what strategies have been recommended to the athlete and by whom, and what strategies they would recommend for other athletes. A combination of open and closed questions was used to provide more detail and elaborate on the closed question answers [20,21]. The questionnaire was comprised of five sections—demographics and sport/training information, current practices, beliefs, evidence, and sources of information.

#### 2.2.1. Demographics and Sport/Training Information

Participants provided basic demographic information including age, gender, race, and country of residence. Additional sport and training information was collected including the participant’s primary sport, the current level of the athlete (i.e., professional, collegiate, recreational), if they followed a training program, had a coach, the number of days per week training, hours per week training, and their top three finishes in the last year.

#### 2.2.2. Closed Questions

The first closed question section asked participants to select what current recovery strategies they used during practice and competition on a range of recovery strategies. Participants were then asked why they currently undertook the specified recovery strategery from a choice of evidence, experience, or both. Subsequently, they were asked to rate their opinion on a range of recovery strategies’ effectiveness. Their belief of the effectiveness was assessed in terms of their perceived benefit of a strategy. A 5-point scale was used of neutral (indifferent/unsure), no effect, minor effect, moderate effect, or major effect. The answers were assigned a numerical value (5 = most benefit, 2 = no benefit, and 1 = neutral). If the athlete rated effectiveness as 4 or 5 then this was coded as a benefit and if they rated a 2 or 3, this was coded as no benefit, and 1 was coded as neutral. The final closed questions asked participants where they received their information regarding recovery strategies. Participants selected all of their sources from a range of people (e.g., coach, health care providers, fellow athletes) to places (e.g., website, research or professional organization, social media, podcasts). The last questions asked participants what recovery strategies have been recommended to them and by whom as well as what strategies they would recommend to other endurance athletes or athletes, and who has asked them for recommendations on recovery strategies along with what they advised.

#### 2.2.3. Open Questions

The open questions allowed participants to provide additional information to the closed questions. First, participants were able to select ’other’ in the current recovery strategies for practice and competition as well as sources of information, and then they were asked to specify. Participants were provided an optional expansion on the limited responses of why the athlete used these recovery strategies (i.e., experience, evidence, or both). The primary open-ended question asked participants to state how they knew they had recovered from training and competition.

### 2.3. Procedure

Before the study began, Institutional Review Board (IRB) approval was confirmed. Participants were recruited and identified from a global endurance athlete population using digital fliers and advertisements sent through several online channels including email, social media groups (i.e., Facebook groups, Twitter, etc.), and online organizations used to facilitate communications between endurance athletes, their teams, and their coaches. Participants consented to participation by e-signing their informed consent prior to starting the online survey. Participants who signed the informed consent form were then instructed to complete an anonymous survey that asked multiple-choice and open-ended questions regarding demographic information, information about their respective sports, and recovery strategies (described above). Participants submitted their survey through Qualtrics upon completion and were thanked for their participation in the research study.

### 2.4. Data Analysis

Responses were reported as a percent of the total for each question. The most common recovery strategies were identified as a total prevalence of ≥15% for training or competition unless smaller distributions were necessary to explain larger themes. Responses that met the ≥15% threshold were analyzed by sex and previous competition outcomes (top three overall/division) using Chi-square (Χ^2^) tests. A ≥ 20% threshold was used for questions regarding information sources and recommendations. Three participants were removed from sex comparisons due to non-response (*n* = 261). Yates corrections for continuity were used for all 2 × 2 analyses. Adjusted standard residuals (ASR) were used if initial Χ*^2^* tests were significant to determine differences from expected for 3 × 2 analyses with a threshold of ±2. An independent samples t-test was used to compare total strategy use by sex and competition outcomes. Mann Whitney U non-parametric tests were used to compare ratings of effectiveness by sex and competition outcomes. Statistical significance was set at *p* < 0.050. Data were analyzed using IBM SPSS version 27 (IBM, Armonk, NY, USA). Open-ended questions were coded in N.Vivo Release 1.5.2 Software (QSR International) into subcategories for subsequent analysis of the frequency of occurrence.

## 3. Results

### 3.1. Participant Characteristics

Participant characteristics for the total sample are presented in Table 1. The majority of the participants were female (52.65%), white (87.5%), and recreational athletes (63.26%).

### 3.2. Use

The use of each individual strategy for training and competition is reported in Table 2. The most common recovery strategies for training and competition included hydration, nutrition, sleep, rest, stretching, foam rolling, active recovery, self-massage, traditional massage, socializing, compression garments, and mindfulness. A ≥ 10% difference between use during training (T) and use during competition (C) was observed for foam rolling (T = 58.9%; C = 48.3%), active recovery (T = 47.1%; C = 34.2%), and self-massage (T = 44.9%; C = 30.8%); all of which were higher for training than competition. Although it did not meet the 10% threshold, hydration was more common after training (Difference = 9.5%) and a traditional massage was more common after a competition (Difference = 9.9%).

Comparisons of strategy use by sex group revealed that more females used massage to recover from training (Χ^2^ = 6.47, *p* = 0.017, ASR = 2.5) and stretching to recover from competition (Χ^2^ = 5.70, *p* = 0.024, ASR = 2.4) compared to males (Figure 1). Although the aforementioned strategies were the only strategies that differed significantly by sex, all of the most common recovery strategies were used to a greater degree by females compared to males for both training and competition. Combined, female athletes reported using significantly more recovery strategies than their male counterparts for training (7.6 ± 3.4 vs. 6.8 ± 3.5; *p* = 0.043, *d* = 0.25, *d*^95%CI^ = 0.08, 0.50) but not for competition (6.7 ± 3.4 vs. 6.1 ± 3.7; *p* = 0.137, *d* = 0.19, *d*^95%CI^ = −0.06, 0.43).

Individuals who reported placing top three overall in competition used significantly more recovery strategies than those who had not placed in the top three overall for both training (8.8 ± 3.9 vs. 6.8 ± 3.2; *p* < 0.001, *d* = 0.61, *d*^95%CI^ = 0.30, 0.91) and competition (7.6 ± 3.5 vs. 6.1 ± 3.5; *p* = 0.008, *d* = 0.41, *d*^95%CI^ = 0.11, 0.71). Likewise, individuals who reported placing top three in their division during competition used significantly more recovery strategies than those who had not placed top three in their division during training (8.3 ± 3.6 vs. 6.4 ± 3.0; *p* < 0.001, *d* = 0.57, *d*^95%CI^ = 0.32, 0.82) and competition (7.3 ± 3.6 vs. 5.8 ± 3.3; *p* < 0.001, *d* = 0.45, *d*^95%CI^ = 0.20, 0.69). Interestingly, a significant inverse relationship was observed between age and total recovery strategies during training (r = −0.21, *p* < 0.001) and competition (r = −0.16, *p* = 0.008) suggesting that increased age was associated with decreased use of recovery strategies. Of the most commonly used recovery strategies, individuals who placed top three overall used foam rolling (Χ^2^ = 6.01, *p* = 0.029, ASR = 2.3), active recovery (Χ^2^ = 6.01, *p* = 0.022, ASR = 2.5), self-massage (Χ^2^ = 4.98, *p* = 0.038, ASR = 2.2), traditional massage (Χ^2^ = 5.86, *p* = 0.025, ASR = 2.4), and socializing (Χ^2^ = 10.22, *p* = 0.003, ASR = 3.4) more than those who did not place top three overall to recover from training (Figure 2). To recover from competition, only socializing (Χ^2^ = 8.63, *p* = 0.032, ASR = 2.3) and compression garments (Χ^2^ = 5.90, *p* = 0.025, ASR = 2.4) were used more by those who placed top three over those who did not. Additionally individuals who placed top three in their division used nutrition (Χ^2^ = 6.26, *p* = 0.019, ASR = 2.5), stretching (Χ^2^ = 5.72, *p* = 0.024, ASR = 2.4), foam rolling (Χ^2^ = 9.57, *p* = 0.002, ASR = 3.1), active recovery (Χ^2^ = 8.20, *p* = 0.006, ASR = 2.9), traditional massage (Χ^2^ = 10.30, *p* = 0.002, ASR = 3.2), socializing (Χ^2^ = 8.35, *p* = 0.006, ASR = 2.9), and compression garments (Χ^2^ = 7.45, *p* = 0.010, ASR = 2.7) more than those who did not place top three in their division. To recover from competition, rest (Χ^2^ = 6.51, *p* = 0.016, ASR = 2.6), foam rolling (Χ^2^ = 5.26, *p* = 0.030, ASR = 2.3), traditional massage (Χ^2^ = 7.99, *p* = 0.007, ASR = 2.8), socializing (Χ^2^ = 9.30, *p* = 0.004, ASR = 3.0) and compression garments (Χ^2^ = 12.39, *p* = 0.001, ASR = 3.5) were used more by those who placed top three in their division compared to those who did not.

### 3.3. Belief

The majority of participants reported that they believed there was a benefit for hydration (T = 97.3%; C = 94.1%), sleep (T = 92.0%; C = 93.4%), rest (T = 91.4%; C = 87.3%), nutrition (T = 91.2%; C = 93.0%), stretching (T = 73.8%; C = 67.4%), foam rolling (T = 65.5%; C = 61.8%), massage (T = 64.4%; C = 61.4%), active recovery (T = 62.1%; C = 59.2%), and self-massage (T = 58.7%; C = 53.6%) for recovery from training. This was not observed for several common strategies including compression garment (T = 28.0%; C = 30.5%), socializing (T = 24.3%; C = 25.8%), and mindfulness (T = 20.0%; C = 18.1). The majority of participants felt neutral regarding the benefits of laser (T = 81.9%; C = 81.1%), ultrasound (T = 77.5%; C = 78.5%), cupping (T = 69.9%; C = 73.4%), dry needling (T = 69.1%; C = 72.6%), contrast bath (T = 65.5%; C = 67.2%), cryotherapy (T = 62.5%; C = 63.6%), electrical stimulation (T = 59.5%; C = 61.7%), imagery (T = 55.2%; C = 59.4%), compressive massage (T = 52.4%; C = 53.0%), and mindfulness (T = 55.2%; C = 52.4%). The majority of participants felt neutral for taping (52.4%) and manipulations (51.1%) for competition only. There were no strategies where the majority of participants reported a belief of no benefit. Use of strategy by belief for the most common recovery strategies is presented in Table 3.

There was a significant effect of belief on use for all strategies (*p* < 0.050) with the exception of hydration for training and sleep for competition. For training, the majority of participants believed that massage had a benefit but only 30.7% of participants used the strategy. For competitions, the majority of participants believed in active recovery, self-massage, and massage, but less than 50% of those who believed in these strategies reported using them. Despite reporting no benefit for hydration, nutrition, rest, and stretching for training, the majority of these individuals reported using these strategies. Similar findings were observed for competition with the exclusion of stretching.

Comparisons of belief by sex revealed that for training, females had a greater belief in the benefit of rest (Χ^2^ = 12.08, *p* = 0.002, ASR = 3.0) and males had a greater disbelief in the benefit of rest (ASR = 3.4). For competition, females had a greater belief in the benefit of massage (Χ^2^ = 10.86, *p* = 0.004, ASR = 3.3) and stretching (Χ^2^ = 10.60, *p* = 0.005, ASR = 3.3) and males had a greater disbelief in the benefit of massage (ASR = 2.1) and stretching (ASR = 2.5). Females had a higher neutral belief in the benefits of massage during competition (ASR = 2.0). Results for belief by sex are reported in Figure 3.

Results for belief by competition outcome are reported in Table 4. Comparisons of belief by competition outcomes revealed that for training, athletes who were top three overall had a greater disbelief in the benefits of massage (Χ^2^ = 8.51, *p* = 0.014, ASR = 2.2), self-massage (Χ^2^ = 9.66, *p* = 0.008, ASR = 2.0), and compression garments (Χ^2^ = 8.26, *p* = 0.016, ASR = 2.9) compared to those who did not place top three overall. Those who did not place top three overall had a higher neutral belief in the benefits of massage (ASR = 2.9) and self-massage (ASR = 2.8). For competition, athletes who placed top three overall had a greater disbelief in the benefits of compression garments (Χ^2^ = 15.26, *p* < 0.001, ASR = 2.0). However, those who placed top three overall had a greater belief in the benefit of self-massage (Χ^2^ = 12.42, *p* = 0.002, ASR = 2.1), compression garments (ASR = 2.1), active recovery (Χ^2^ = 7.81, *p* = 0.020, ASR = 2.0), and mindfulness (Χ^2^ = 6.64, *p* = 0.036, ASR = 2.1) during competition. Those who did not place top three overall had a higher neutral belief in the benefits of massage (Χ^2^ = 12.16, *p* = 0.002, ASR = 3.3), self-massage (ASR = 3.5), compression garments (ASR = 3.9), active recovery (ASR = 2.7), and mindfulness (ASR = 2.3).

Comparisons of belief by competition outcomes revealed that for training, the top three by division had a greater disbelief in the benefits of foam rolling (Χ^2^ = 8.85, *p* = 0.012, ASR = 2.3) compared to those who did not place top three overall. Those who did not place top three in their division had a higher neutral belief in the benefits of self-massage (Χ^2^ = 7.77, *p* = 0.021, ASR = 2.7), compression garments (Χ^2^ = 7.97, *p* = 0.019, ASR = 2.7), active recovery (Χ^2^ = 10.21, *p* = 0.006, ASR = 2.9), and foam rolling (ASR = 2.3). For competition, athletes who placed top three in their division had a greater disbelief in the benefits of massage (Χ^2^ = 16.61, *p* < 0.001, ASR = 2.0), stretching (Χ^2^ = 13.76, *p* < 0.001, ASR = 2.7), and foam rolling (Χ^2^ = 14.82, *p* < 0.001, ASR = 3.0). Those who did not place top three overall had a higher neutral belief in the benefits of massage (ASR = 3.9), self-massage (Χ^2^ = 12.99, *p* = 0.002, ASR = 3.6), compression garments (Χ^2^ = 15.57, *p* < 0.001, ASR = 3.9), active recovery (Χ^2^ = 9.81, *p* = 0.007, ASR = 3.1), stretching (ASR = 3.0), foam rolling (ASR = 3.1), and socializing (Χ^2^ = 7.45, *p* = 0.024, ASR = 2.5). However, those who placed top three overall had a greater belief in the benefit of self-massage (ASR = 2.2), compression garments (ASR = 2.6), active recovery (ASR = 2.0), and socializing (ASR = 2.2) during competition.

### 3.4. Ratings of Effectiveness

Means and 95%CI for ratings of effectiveness by sex and top three placement in division for training and competition are presented in Figure 4 and Figure 5. Strategies with a median effectiveness rating of 5 (major effect) for training and competition included sleep, nutrition, hydration, and rest. Strategies with a median effectiveness rating of 4 (moderate effect) for training and competition included massage, self-massage, active recovery, stretching, and foam rolling. A median score of 3 (neutral effect) was observed for all remaining strategies for both training and competition. For training, results of Mann-Whitney U tests examining ratings of effectiveness for training revealed that female athletes had a significantly higher mean ranking for cryotherapy (*p* = 0.005), contrast bath (*p* = 0.038), rest (*p* < 0.001), massage (*p* = 0.023), active recovery (*p* = 0.024), stretching (*p* = 0.035), foam rolling (*p* = 0.026), cupping (*p* = 0.002), dry needling (*p* = 0.007), and socializing (*p* = 0.013). For competition, females had had a significantly higher mean ranking for hydration (*p* = 0.003), heat (*p* = 0.039), ice bath (*p* < 0.001), massage (*p* = 0.003), stretching (*p* = 0.010), cupping (*p* = 0.009), dry needling (*p* = 0.038), and socializing (*p* = 0.019).

Further, athletes who were top three overall in their competition had a significantly higher mean ranking for active recovery following competition (*p* = 0.034) compared to those who did not place top three overall while those who were not top three overall had a significantly higher mean ranking for ice baths following training (*p* = 0.029). Athletes who placed top three in their division had a significantly higher mean ranking for sleep (*p* = 0.040), nutrition (*p* = 0.029), compressive massage (*p* = 0.034), and active recovery (*p* = 0.019) following competition and for nutrition (*p* = 0.011) and massage (*p* = 0.036) following training. Athletes who were not top three in their division had a significantly higher mean ranking for contrast baths (*p* = 0.011) compared to those who did place top three in their division.

### 3.5. Sources of Information and Recommendations

On average, athletes received their information on recovery strategies from 2.8 ± 2.3 sources which did not differ across groups. The most common sources (≥20% prevalence) of recovery strategy information included a coach (48.3%), a fellow athlete (47.5%), a website (32.7%), a physical therapist (26.2%), a research article (22.4%), and social media (20.9%). It was more common for female athletes (56.8%) to receive their information from coaches compared to males (males = 37.7%, Χ^2^ = 9.53, *p* = 0.003, ASR = 3.1). There were no other differences by sex or by competition outcomes.

On average, athletes reported being recommended 10.2 ± 6.6 recovery strategies. Female athletes (11.0 ± 6.1) had a significantly higher number of strategies recommended to them compared to male athletes (9.2 ± 7.1; *p* = 0.033, *d* = 0.27. *d*^95%CI^ = 0.02, 0.51). Additionally, athletes who placed top three overall (12.2 ± 6.7) had a significantly higher number of strategies recommended to them compared to athletes who did not place top three overall (9.7 ± 6.5; *p* = 0.013, *d* = 0.38. *d*^95%CI^ = 0.08, 0.68). Finally, athletes who placed top three in their division (12.0 ± 6.6), had a significantly higher number of strategies recommended to them compared to athletes who did not place top three in their division (8.8 ± 6.3; *p* < 0.001, *d* = 0.50. *d*^95%CI^ = 0.26, 0.75).

The most commonly recommended recovery strategies included hydration (81.8%), nutrition (81.1%), sleep (77%), stretching (74.2%), foam rolling (71.2%), rest (70.1%), massage (62.5%), active recovery (50.0%), self-massage (47.7%), and ice bath (43.9%). It was most common for female athletes to be recommended foam-rolling (females = 80.6%, males = 60.7%; Χ^2^ = 12.59, *p* < 0.001, ASR = 3.5), rest (females = 76.3%, males = 63.1%; Χ^2^ = 5.36, *p* = 0.029, ASR = 2.3), traditional massage (females = 69.1%, males = 54.9%; Χ^2^ = 5.55, *p* = 0.026, ASR = 2.4), self-massage (females = 54.0%, males = 41.0%; Χ^2^ = 4.38, *p* = 0.049, ASR = 2.1), ice baths (females = 53.2%, males = 32.8%; Χ^2^ = 11.05, *p* = 0.001, ASR = 3.3), taping (females = 40.3%, males = 27.9%; Χ^2^ = 4.44, *p* = 0.050, ASR = 2.1) and dry-needling (females = 26.6%, males = 14.8%; Χ^2^ = 5.50, *p* = 0.028, ASR = 2.3).

Athletes who placed top three overall were more likely to be recommended massage (Top3 = 80.0%, NotTop3 = 57.7%; Χ^2^ = 9.22, *p* = 0.004, ASR = 3.0), compression garments (Top3 = 50.9%, NotTop3 = 32.2%; Χ^2^ = 6.59, *p* = 0.016, ASR = 2.6), and manipulation (Top3 = 50.9%, NotTop3 = 28.8%; Χ^2^ = 9.51, *p* = 0.003, ASR = 3.1). Athletes who placed top three in their division were more likely to be recommended ice baths (Top3 = 52.2%, NotTop3 = 37.7%; Χ^2^ = 5.49, *p* = 0.027, ASR = 2.3), massage (Top3 = 79.6%, NotTop3 = 49.7%; Χ^2^ = 24.78, *p* < 0.001, ASR = 5.0), self-massage (Top3 = 61.9%, NotTop3 = 37.1%; Χ^2^ = 16.01, *p* < 0.001, ASR = 4.0), compressive massage (Top3 = 41.6%, NotTop3 = 22.5%; Χ^2^ = 11.06, *p* = 0.001, ASR = 3.3), compression garments (Top3 = 47.8%, NotTop3 = 27.2%; Χ^2^ = 11.95, *p* = 0.001, ASR = 3.5), electrical stimulation (Top3 = 28.3%, NotTop3 = 17.2%; Χ^2^ = 4.65, *p* = 0.045, ASR = 2.2), active recovery (Top3 = 61.9%, NotTop3 = 41.1%; Χ^2^ = 11.28, *p* = 0.001, ASR = 3.4), foam rolling (Top3 = 81.4%, NotTop3 = 63.6%; Χ^2^ = 10.03, *p* = 0.002, ASR = 3.2), dry needling (Top3 = 30.1%, NotTop3 = 14.6%; Χ^2^ = 9.314, *p* = 0.004, ASR = 3.1), manipulations (Top3 = 46.0%, NotTop3 = 23.8%; Χ^2^ = 14.30, *p* < 0.001, ASR = 3.8), and taping (Top3 = 41.6%, NotTop3 = 29.1%; Χ^2^ = 4.44, *p* = 0.048, ASR = 2.1).

On average, athletes reported being recommended 3.9 ± 3.1 recovery strategies with no differences by groups. These strategies were most commonly recommended by a fellow athlete (62.1%), a coach (54.9%), a physical therapist (34.1%), a website (32.6%), an athletic trainer (24.2%), social media (23.1%), a chiropractor (22.7%), or a research article (20.1%). Athletes who placed top three overall were more likely to be recommended recovery strategies by athletic trainers (Top3 = 36.4%, NotTop3 = 21.2%; Χ^2^ = 5.47, *p* = 0.031, ASR = 2.3), and chiropractors (Top3 = 34.5%, NotTop3 = 19.7%; Χ^2^ = 5.44, *p* = 0.031, ASR = 2.3).

### 3.6. Assessment of Recovery

When analyzing the open-ended questions on how the athlete knew they had recovered from practice and competition, participants primarily relied on subjective feelings of how their body felt, soreness, energy levels, wanting to train again along with their heart rate to determine if they had recovered (Figure 6). For recovery from a competition, participants stated that they have recovered when they wanted to sign-up for another competition or were ready for another competition.

## 4. Discussion

This study was aimed at establishing current practices, attitudes, beliefs, and sources of information toward recovery strategies in endurance athletes. When examining recovery practices, endurance athletes in the current study favored lifestyle choice recovery practices including hydration, nutrition, sleep, and rest in terms of use, belief, and effectiveness. These results are similar to Venter [22] who reported that hydration and sleep were rated as an important recovery strategy for all team athletes regardless of gender or level of participation. Even if participants reported no benefit in these recovery strategies, the majority of these individuals still used them. Hydration is the most reported recovery strategy utilized and found as beneficial, and it is rated equally as effective in both training and competition settings. Hydration was used more during training than in competition, although it did not meet the 10% threshold. Long endurance sessions impact hydration status, especially in a hot environment, and rehydration is an important aspect of the recovery process [23]. Therefore, it is important for these athletes to replenish their fluid loss in both training and competition for adequate recovery. The goal of nutrition is to maximize the functional metabolic adaptation to a periodized training program, support the athlete to stay healthy and injury-free [24]. A crucial part of the stress-recovery balance is the management of an athlete’s sleep [22,25,26]. Endurance athletes value sleep as a recovery method which is similar to previous research on adult athletes [10,22]. Sleep loss can lead to decreases in physical output and affect psychological performance elements such as decision making and mood states [27].

Comparing differences in the use of recovery strategies between training and competition, there were ≥10% differences in foam rolling, active recovery, and self-message with all of these strategies being utilized more in training than in competition. These recovery strategies may be applied more in training in order to minimize any negative impacts from the session to achieve their performance goals [28]. Improper program design (i.e., too much volume, too much intensity, etc.) can potentially lead to under-recovery, overtraining, and/or injury [9]. With increasing training load, recovery demands increase proportionally. Massage, however, was utilized more after a competition. Massage may be employed more after competition due to the costs and time associated with it [29]. Furthermore, massages are frequently offered at the competition site providing easy access for endurance athletes.

Previous research has demonstrated that endurance athletes commonly use lower extremity compression garments especially runners [30]. However, this was not the case in the current study. Athletes who wear compression garments for recovery report perceived faster recovery and improvements in sports performance [30]. In the current study, athletes value compression garments more in competition than in practice. This could be due to differences in physiological demands. Furthermore, the athletes in the current study may be utilizing the compressions garments to prevent injuries or reduce symptoms of an injury [30].

Despite the popularity of ice baths/CWI and cryotherapy, this was not a popular recovery strategy or perceived to be beneficial by the participants. This is contrary to Division I athletes [10] and international team sport athletes [18,22] who utilize CWI as a common recovery strategy. There is still a lack of consistency in the literature regarding the efficacy of these recovery strategies. In marathon runners, whole body cryotherapy was harmful to recovery of muscle function compared to CWI and both recovery strategies were no more effective than a placebo [31]. Recently, Malta and colleagues [32] reported that regular use of CWI has detrimental effects on resistance training but it does not impact aerobic exercise performance. Since these athletes are relying on evidence as well as experience for using recovery strategies along with almost a quarter of participants receiving information from scientific journal articles, they may know of the current literature.

Regardless that recovery is a multidimensional process, few endurance athletes utilize psychological skills (i.e., relaxation techniques, mindfulness, or imagery) to recover from practice and competition. The development of psychological strategies is important to assist in counteracting stress, increasing coping strategies, decreasing anxiety, and increasing motivation which in turn enhances recovery and subsequently performance [33].

Communication with teammates, coaches, family, and friends can aid in psychological recovery. Socialization as a recovery strategery was used more in competition by the top three overall finishers and top three in their division. Participants who placed in the top three overall also had a greater belief in socialization compared to those who did not place in the top three. According to Venter [22], communicating with friends is perceived as one of the most important recovery modalities. Social networks allow athletes to process problems, disappointments, joys, and stresses of life [34].

In general, female endurance athletes and top three finishers overall or in their division use more recovery strategies. Females significantly utilized massage and stretching more to recover from competition compared to males. Furthermore, females who use all the most common recovery strategies reported more compared to males in both training and competition, however, it was non-significant. Endurance female athletes in the current study are not performing the recommended recovery strategies according to research including cooling recovery methods (i.e., CWI) and active recovery [11]. When examining skill levels, individuals who reported placing top three overall or top three in their division used significantly more recovery strategies than those who had not placed top three in competition, similar to past research [16].

There are discrepancies between the beliefs and practices of recovery strategies with endurance athletes in terms of believing in massage after training and competition and believing in active recovery and self-massage after competition but not utilizing these strategies. Furthermore, females report believing in massage more than males. The top three finishers overall had greater disbelief in massage and self-massage after training, however, they had a greater belief in self-massage after a competition. Additionally, the top three finishers overall had a greater belief in active recovery following a competition. In elite ultra-marathon runners, massage significantly improved perceived pain [35]. The costs and time associated with massage or self-massage might be too high despite the benefits. Active recovery significantly improves delayed onset muscle soreness (DOMS) but it does not impact perceived fatigue [36]. Some of these athletes may have time constraints and might select quicker strategies despite the benefits of active recovery.

Most endurance athletes are relying on the people around them and websites for information and recommendations on recovery. This may impact and influence the recovery strategy choices. Almost half of participants receive information from a coach or fellow athlete. More specifically, females receive their recovery information from coaches more than males. Previous research has demonstrated that the choice of the recovery strategy is influenced by what the coaches and support staff prefer [37]. Furthermore, coaches prefer and mostly acquire their coaching knowledge from informal learning activities and self-directed learning with other coaches and colleagues which may impact their recommendations for recovery strategies [38].

Moreover, athletes are discussing recovery strategies with each other, frequently recommending hydration, nutrition, and sleep which aligns with the present findings of unitization and benefits of these strategies. Coaches and fellow athletes could all potentially be comfortable and easily accessible sources of information [39]. Athletes also model the behaviors of what they observe in the media by elite-level athletes, who may have sponsorships with certain recovery strategies which could be portrayed and/or advertised on websites. The popularity of websites is in line with previous research on nutritional sources of information [40,41,42]. However, information on websites is not always reliable. Therefore, proper education is needed.

Recovery from practice and competition was mainly determined by self-perceptions of how they felt. Monitoring the training load and the athlete’s response (e.g., fitness, fatigue, and performance) is critical in making informed decisions on training and recovery [5,6]. Saw et al. [43] reported that subjective measures compared to objective measures reflect acute and chronic training loads with superior sensitivity and consistency when truthfully reported. More specifically, subjective well-being is typically impaired with acute increases in training load as well as chronic training. Only a little over 6% of endurance athletes used their heart rate (HR) measurements as an indication for recovery. HR measurements provide information into fitness-related changes to training and are non-invasive, time-efficient, and relatively cheap [44]. Combining subjective and objective measures provides a clearer interpretation of the recovery status.

### Limitations and Future Research

It is important to consider some of the limitations within the present study. Some caution should be taken when interpreting the results of the study because data were collected during the COVID-19 pandemic and no in-person competitions were occurring. Therefore, some participants may not recall or did not report the strategies utilized to recover from a competition. Additionally, training and work activities may have been impacted subsequently changing their current recovery methods. The current study examined several different categories of endurance athletes. Therefore, future research could consider investigating one type of endurance athlete (i.e., recreational marathon runners). Several of the participants are either current or former collegiate or professional athletes which may impact the results. The survey was distributed via email. As a result of the process, the participants self-selected into the study by volunteering to respond to the survey. The lack of random selection is a threat to external validity and further limits the generalizability of the current study. Additionally, the measures for this study were all questionnaires. Questionnaires rely on the self-report of the participants and the truthfulness of their responses. Some participants may have not been familiar with the terminology of the recovery strategery and may use a different term. Incorporating a more structured interview style could avoid any misunderstandings around the questions and allow more detailed responses. Future research should examine the preferential use of a particular recovery strategy along with when and how frequently the strategy is utilized. Additionally, future research should consider the physiological and psychological demands of the participants and develop a multimodal recovery intervention study. Finally, athletes’ injuries should be considered as this may impact their recovery practices.

## 5. Conclusions

This study described endurance athletes’ recovery practices and highlights differences between sex and top three finishers overall or in their division. Endurance athletes are primarily utilizing, believing in, and rate effectiveness of lifestyle recovery practices (i.e., hydration, nutrition, and sleep). These athletes primarily use these strategies based on both evidence and experience while receiving their information on recovery from a coach or fellow athlete. Recovery from practice and competition was mainly determined by self-perceptions of how they felt.

These findings are useful for healthcare professionals, practitioners, and coaches in understanding recovery practices among endurance athletes. These results suggest that endurance athletes and coaches should be educated on the benefits of different modes of recovery, specifically psychological and emotional recovery. It is vital to educate those who are closest to the athletes.

## Figures and Tables

**Figure 1 jfmk-07-00022-f001:**
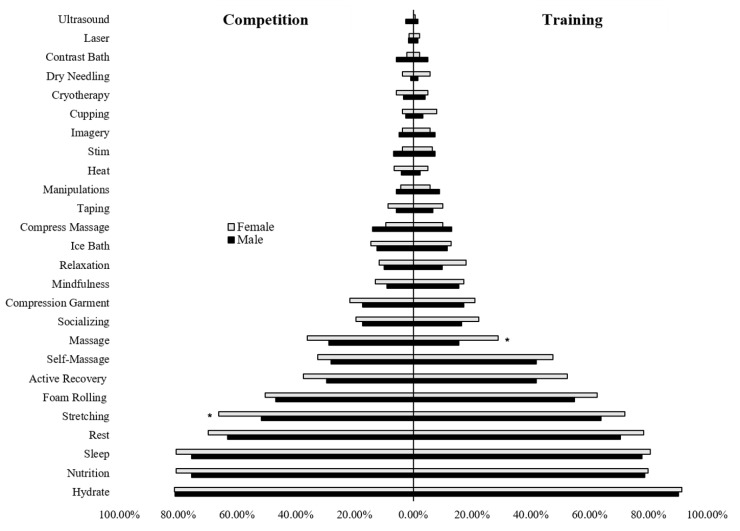
Recovery strategy use in training and competition by sex. Note. * = *p* < 0.05.

**Figure 2 jfmk-07-00022-f002:**
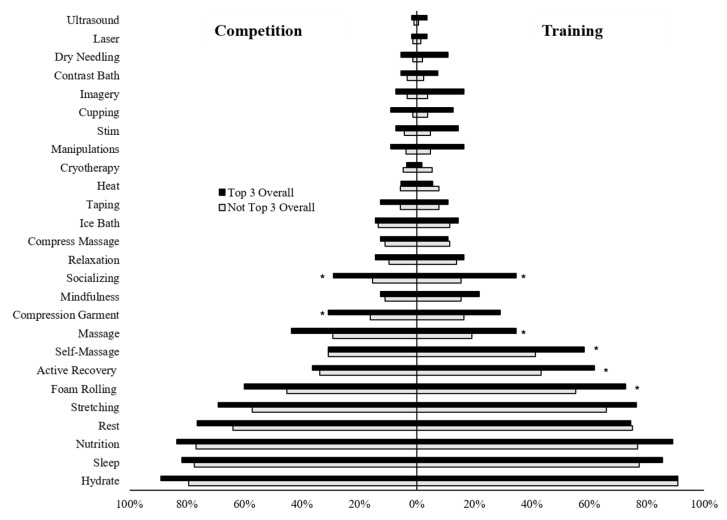
Recovery strategy used by the top three overall in a competition. Note. * = *p* < 0.05.

**Figure 3 jfmk-07-00022-f003:**
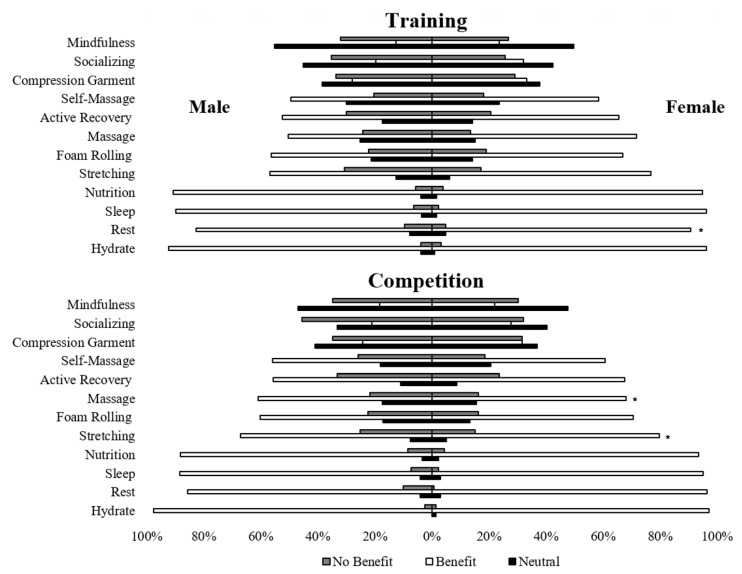
Belief in the recovery strategy in training and competition by sex. Note. * = Females had greater belief in the benefit and males had a greater disbelief in the benefit (α = 0.05).

**Figure 4 jfmk-07-00022-f004:**
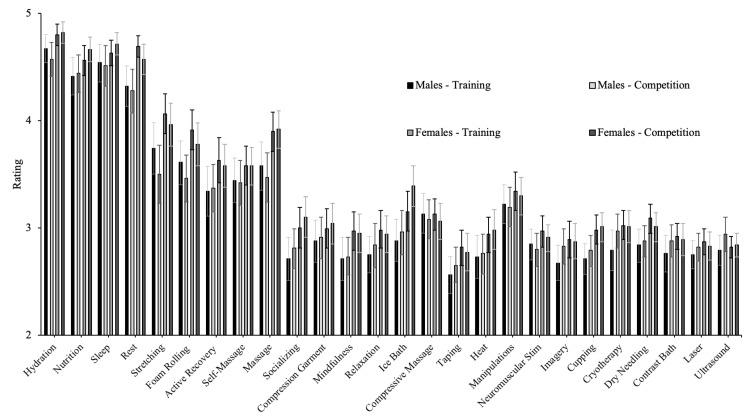
Ratings of recovery strategy effectiveness by sex. For training, female athletes had a significantly higher mean ranking for cryotherapy, contrast bath, rest, massage, active recovery, stretching, foam rolling, cupping, dry needling, and socializing. For competitions, females had had a significantly higher mean ranking for hydration heat, ice bath, massage, stretching, cupping, dry needling, and socializing.

**Figure 5 jfmk-07-00022-f005:**
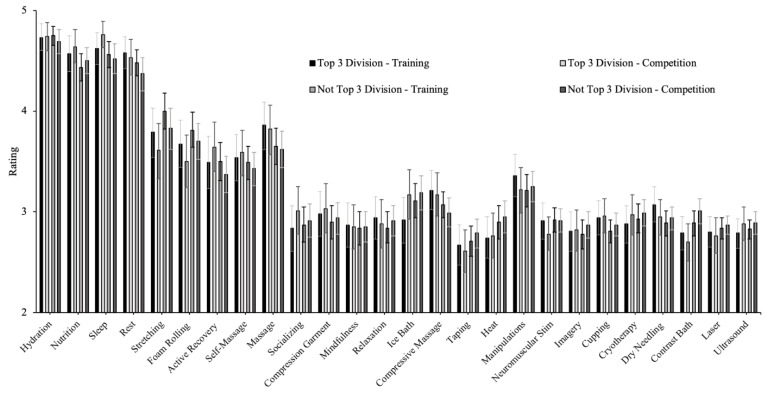
Ratings of recovery strategy effectiveness by competition outcomes—top three placement in the division. Placing the top three in the division had a significantly higher mean ranking for sleep, nutrition, compressive massage, and active recovery following competition and for nutrition and massage following training. Athletes who were not top three in their division had a significantly higher mean ranking for contrast baths.

**Figure 6 jfmk-07-00022-f006:**
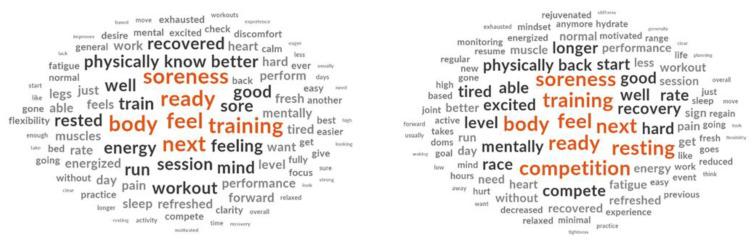
How endurance athletes know they have recovered from practice (**left**) and competition (**right**). This word cloud is based on the frequency of responses to the open questions. The more times a word is used the larger it appears and changes color.

**Table 1 jfmk-07-00022-t001:** Participant characteristics.

	M ± SD or *n* (%)		
	Total (*n* = 264)	Male (*n* = 122)	Female (*n* = 139)
**Age (y)**	39.82 ± 13.84	43.2 ± 14.28	36.58 ± 12.78
**Race**			
American Indian or Alaskan Native	4 (1.5)	2 (1.6)	2 (1.4)
Asian	13 (4.9)	3 (2.5)	10 (7.2)
Black or African American	4 (1.5)	1 (0.8)	3 (2.2)
Native Hawaiian or Pacifica Islander	1 (0.4)	0	1 (0.7)
White	231 (87.5)	111 (91.0)	118 (84.9)
Multiracial	7 (2.7)	4 (3.3)	3 (2.2)
Not reported	4 (1.5)	1 (0.8)	2 (1.4)
**Ethnicity**			
Hispanic or Latino	27 (10.5)	13 (11.0)	14 (10.3)
**Sport**			
Cycling	55 (20.8)	42 (34.4)	13 (9.4)
Paracycling	1 (0.4)	0	1 (0.7)
Nordic skiing	1 (0.4)	1 (0.8)	0
Running	99 (37.5)	29 (23.8)	68 (48.9)
Race walking	1 (0.4)	0	1 (0.7)
Rowing	8 (3.0)	4 (3.3)	4 (2.9)
Swimming	7 (2.7)	3 (2.5)	4 (2.9)
Triathlon	89 (33.7)	42 (34.4)	46 (33.1)
Wheelchair racing	1 (0.4)	0	1 (0.7)
Snowshoeing	1 (0.4)	0	1 (0.7)
Aquabiking	1 (0.4)	1 (0.8)	
**Competition Level**			
Professional	14 (5.3)	7 (5.7)	7 (5.0)
Collegiate Athlete	41 (15.5)	12 (9.8)	29 (20.9)
Former Collegiate Athlete	42 (15.9)	21 (17.2)	21 (15.1)
**Placing**			
Top 3 Overall	55 (20.8)	24 (19.7)	31 (22.3)
Top 3 Division	113 (42.8)	42(34.4)	70 (50.4) *

* = distribution significantly higher than expected at *p* < 0.05.

**Table 2 jfmk-07-00022-t002:** Use of recovery strategy (%).

Recovery Strategry	Training	Competition
Hydration	90.9%	81.4%
Nutrition	79.5% ^‡^	78.3%
Sleep	79.1%	78.3%
Rest	74.9%	66.5% ^‡^
Stretching	68.1% ^‡^	59.7% *
Foam Rolling	58.9% ^†,‡^	48.3% ^‡^
Active Recovery	47.1% ^†,‡^	34.2%
Self-Massage	44.9% ^†^	30.8%
Massage	22.4% *^,†,‡^	32.3% ^‡^
Socializing	19.4% ^†,‡^	18.3% ^†,‡^
Compression Garment	19.0% ^†,‡^	19.4% ^†,‡^
Mindfulness	16.7%	11.4%
Relaxation	14.4%	10.6%
Ice Bath	12.2%	13.7%
Compress Massage	11.4%	11.4%
Taping	8.4%	7.2%
Heat	7.2%	5.7%
Manipulations	7.2%	4.9%
Stim	6.8%	4.9%
Imagery	6.5%	4.2%
Cupping	5.7%	3.0%
Cryotherapy	4.6%	4.6%
Dry Needling	3.8%	2.3%
Contrast Bath	3.4%	3.8%
Laser	1.9%	1.5%
Ultrasound	1.1%	1.1%

* = Significantly different by sex (α = 0.05); ^†^ = Significantly different by top three overall (α = 0.05); ^‡^ = Significantly different by top three in division (α = 0.05).

**Table 3 jfmk-07-00022-t003:** Use of recovery strategy by belief in the strategy.

	Training				Competition			
	Neutral (N)	Benefit (B)	No Benefit (NB)		Neutral	Benefit	No Benefit	
Hydrate	50.0%	92.1%	80.0%	*p* = 0.066	40.0% ^‡^	86.2% ^†^	66.7%	*p* = 0.006
Nutrition	42.9% ^‡^	82.8% ^†^	56.3% ^‡^	*p* = 0.002	50.0% ^‡^	83.1% ^†^	63.6% ^‡^	*p* = 0.036
Sleep	77.8%	81.9% ^†^	25.0% ^‡^	*p* < 0.001	66.7%	81.5%	60.0%	*p* = 0.173
Rest	33.3% ^‡^	77.9% ^†^	53.8%	*p* = 0.002	35.7% ^‡^	72.8% ^†^	56.3%	*p* = 0.007
Stretching	18.8% ^‡^	75.5% ^†^	61.5%	*p* < 0.001	19.0% ^‡^	78.0% ^†^	35.7% ^‡^	*p* < 0.001
Foam Rolling	10.5% ^‡^	76.6% ^†^	48.0% ^‡^	*p* < 0.001	10.0% ^‡^	68.8% ^†^	34.7% ^‡^	*p* < 0.001
Active Recovery	0.0% ^‡^	61.0% ^†^	38.9%	*p* < 0.001	13.5% ^‡^	47.8% ^†^	20.7% ^‡^	*p* < 0.001
Self-Massage	6.1% ^‡^	65.1% ^†^	32.1% ^‡^	*p* < 0.001	1.6% ^‡^	49.6% ^†^	31.1%	*p* < 0.001
Massage	2.4% ^‡^	30.7% ^†^	14.6%	*p* < 0.001	2.1% ^‡^	46.9% ^†^	25.6%	*p* < 0.001
Socializing	5.4% ^‡^	44.3% ^†^	18.6%	*p* < 0.001	6.0% ^‡^	41.7% ^†^	20.0%	*p* < 0.001
Compression Garment	1.0% ^‡^	47.9% ^†^	16.7%	*p* < 0.001	3.3% ^‡^	43.7% ^†^	17.8%	*p* < 0.001
Mindfulness	0.8% ^‡^	56.0% ^†^	18.5%	*p* < 0.001	0.0% ^‡^	35.7% ^†^	16.4%	*p* < 0.001

^†^ = Adjusted standard residual > 2.0; ^‡^ = Adjusted standard residual > −2.0.

**Table 4 jfmk-07-00022-t004:** Belief by competition outcome.

	Training						Competition					
	Top 3 Division			Not Top 3 Division			Top 3 Division			Not Top 3 Division		
	N	B	NB	N	B	NB	N	B	NB	N	B	NB
Sleep	5.3%	91.2%	3.5%	2.0%	92.7%	5.3%	1.9%	94.4%	3.7%	2.9%	92.6%	4.4%
Nutrition	3.6%	91.1%	5.4%	2.0%	91.3%	6.7%	0.9%	92.5%	6.5%	3.7%	93.3%	3.0%
Hydrate	1.8%	96.4%	1.8%	0.0%	98.0%	2.0%	0.0%	96.2%	3.8%	3.8%	92.5%	3.8%
Rest	3.6%	91.9%	4.5%	3.4%	91.1%	5.5%	1.9%	91.4%	6.7%	9.2%	84.0%	6.9%
Massage	10.8%	69.4%	19.8%	21.1%	60.6%	18.3%	8.7%	67.3%	24.0%	29.5%	56.6%	14.0%
Self-Massage	11.7%	62.2%	26.1%	25.2%	55.9%	18.9%	15.4%	61.5%	23.1%	36.4%	47.3%	16.3%
Compression Garment	29.5%	33.9%	36.6%	46.5%	23.2%	30.3%	24.8%	39.0%	36.2%	49.6%	23.3%	27.1%
Active Recovery	3.6%	62.5%	33.9%	14.6%	61.8%	23.6%	7.6%	65.7%	26.7%	22.7%	53.9%	23.4%
Stretching	3.5%	70.8%	25.7%	8.2%	76.2%	15.6%	2.8%	65.1%	32.1%	13.8%	69.2%	16.9%
Foam Rolling	9.0%	64.9%	26.1%	19.4%	66.0%	14.6%	8.7%	61.5%	29.8%	24.0%	62.0%	14.0%
Mindfulness	44.4%	20.4%	35.2%	50.0%	19.7%	30.3%	46.0%	22.0%	32.0%	57.4%	15.5%	27.1%
Socializing	32.1%	27.5%	40.4%	40.8%	21.8%	37.3%	34.3%	33.3%	32.4%	50.8%	20.3%	28.9%
	**Top 3 Overall**			**Not Top 3 Overall**			**Top 3 Overall**			**Not Top 3 Overall**		
	**N**	**B**	**NB**	**N**	**B**	**NB**	**N**	**B**	**NB**	**N**	**B**	**NB**
Sleep	3.6%	90.9%	5.5%	3.4%	92.3%	4.3%	1.9%	92.5%	5.7%	2.6%	93.7%	3.7%
Nutrition	1.9%	90.7%	7.4%	2.9%	91.3%	5.8%	1.9%	94.3%	3.8%	2.7%	92.6%	4.8%
Hydrate	1.9%	94.4%	3.7%	0.5%	98.1%	1.5%	0.0%	96.2%	3.8%	2.7%	93.5%	3.8%
Rest	1.9%	94.3%	3.8%	3.9%	90.6%	5.4%	1.9%	90.6%	7.5%	7.1%	86.3%	6.6%
Massage	3.7%	72.2%	24.1% ^†^	20.2% ^†^	62.1%	17.7%	3.8%	69.2%	26.9%	25.0% ^†^	58.9%	16.1%
Self-Massage	5.7%	62.3%	32.1% ^†^	23.0% ^†^	57.5%	19.5%	7.8%	66.7% ^†^	25.5%	32.6% ^†^	50.3%	17.1%
Compression Garment	22.2%	35.2%	42.6% ^†^	43.7%	26.1%	30.2%	15.4%	42.3% ^†^	42.3% ^†^	45.3% ^†^	27.1%	27.6%
Active Recovery	1.9%	66.7%	31.5%	11.9%	60.7%	27.4%	3.8%	71.2% ^†^	25.0%	19.4% ^†^	55.6%	25.0%
Stretching	1.8%	72.7%	25.5%	7.4%	74.0%	18.6%	1.9%	66.0%	32.1%	11.0%	67.6%	21.4%
Foam Rolling	7.5%	66.0%	26.4%	16.9%	65.2%	17.9%	7.8%	64.7%	27.5%	19.9%	60.8%	19.3%
Mindfulness	40.4%	23.1%	36.5%	49.7%	18.8%	31.5%	38.0%	28.0% ^†^	34.0%	56.7% ^†^	15.2%	28.1%
Socializing	29.4%	33.3%	37.3%	38.7%	22.1%	39.2%	30.6%	32.7%	36.7%	46.7%	24.4%	28.9%

^†^ = Adjusted standard residual > 2.0; Note: N = Neutral; B = Benefit; NB = No Benefit.

## Data Availability

The data presented in this study are available on request from the corresponding author.

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
