# Peer review of "Recovery Strategies in Endurance Athletes"

_jfmk, 2022, doi:10.3390/jfmk7010022_

Round 1

Reviewer 1 Report

Congratulations on this manuscript. In my opinion, the authors did a great job and the manuscript is easy to read and understand, relevant for coaches, athletes and further research on this topic. Nevertheless, some specific comments are provided below.

Abstract:

- Abstract should include more specific results, please.

Introduction

- I am not sure if endurance athletes are a significant proportion of the population….and 4 million participants in triathlons, depending on other issues. For instance, some are repeated from competition to competition, perhaps. And other issues. Is it possible to change or rewrite the first paragraph? I think that are other forms to highlight the importance of endurance.

- The sentence “ The prioritization on high-level performance often leads to an imbalance between stress and recovery” is not clear. It should be supported by literature and the most important balance should be between load and recovery. The consequence of load is some acute changes that need to recover to elevate performance. The authors should think to rewrite this part and consider using some other concepts properly.

- Paragraphs 50-56 could be included in the previous or following paragraph.

Material and Methods:

-Limitations: Different male and female numbers; a large number of former athletes. Could these factors influence results? This should be included in the discussion/limitations.

-How did the authors guarantee anonymity?

Results:

-The authors did a good job in data treatment. Nevertheless, is there any advantage to comparing females vs males? Considering the different numbers of males and females, this is a limitation for me.

 Discussion:

-Very good work and friendly reader discussion. Nevertheless, I suggest that authors should include the different numbers of females and males as a limitation of the study. The study must be interpreted considering this.

Reviewer 2 Report

Braun-Trocchio and colleagues in their article entitled “Recovery Strategies in Endurance Athletes” examined the current implementation, beliefs and sources of information associated with recovery strategies in endurance athletes.

Overall, the authors have done a good job, albeit with some difficulties as they point out in the section on limitations.

I believe the work is suitable for publication after minor revision of the following points:

  1. Introduction: lines 57-72 contain a list of information that is not discussed and not very detailed. I suggest the authors to rework this part, for example, by better discussing the effects of gender on recovery strategies.
  1. Materials and Methods: In section 2.1. “Participants”, I found an inconsistency between the total number of participants enrolled in the study and the division into individual groups. How many subjects are enrolled? Please also clarify this point in the abstract. 
  1. Results: Before section 3.1. “Use”, I believe that the authors should include a section containing all information about the enrolled participants and their division into groups. In addition, the authors could add a table summarizing these characteristics (e.g. number of participants, gender, average age, membership category, type of sport practiced). Furthermore, in lines 190-192, the authors should provide some additional information regarding the result obtained. 
  1. Results – section 3.3. “Rate of Effectiveness”: The authors do not mention whether there is statistical significance between the results obtained. 
  1. Figure captions. The captions of Figures 1, 2, 3, 4 and 5 should be implemented with a brief description of the results obtained. 
  1. Discussion. I think that section 4 “Discussion” is really too long. The authors should make an effort to rework it, discussing their data and providing evidence to support or not support the results obtained.
